# Coffee Ingestion Improves 5 km Cycling Performance in Men and Women by a Similar Magnitude

**DOI:** 10.3390/nu11112575

**Published:** 2019-10-25

**Authors:** Neil D. Clarke, Nicholas A. Kirwan, Darren L. Richardson

**Affiliations:** School of Life Sciences, Faculty of Health and Life Sciences, Coventry University, Coventry CV12DS, UK; kirwann@cucollege.coventry.ac.uk (N.A.K.); ad1534@coventry.ac.uk (D.L.R.)

**Keywords:** caffeine, ergogenic aid, time trial, afferent responses, sex differences

## Abstract

Caffeine is a well-established ergogenic aid, although research to date has predominantly focused on anhydrous caffeine, and in men. The primary aim of the present study was to investigate the effect of coffee ingestion on 5 km cycling time trial performance, and to establish whether sex differences exist. A total of 38 participants (19 men and 19 women) completed a 5 km time trial following the ingestion of 0.09 g·kg^-1^ coffee providing 3 mg·kg^-1^ of caffeine (COF), a placebo (PLA), in 300 mL of water, or control (CON). Coffee ingestion significantly increased salivary caffeine levels (*p* < 0.001; $\eta_{P}^{2}$ = 0.75) and, overall, resulted in improved 5 km time trial performance (*p <* 0.001; $\eta_{P}^{2}$ = 0.23). Performance following COF (482 ± 51 s) was faster than PLA (491 ± 53 s; *p* = 0.002; *d* = 0.17) and CON (487 ± 52 s; *p* =0.002; *d* = 0.10) trials, with men and women both improving by approximately 9 seconds and 6 seconds following coffee ingestion compared with placebo and control, respectively. However, no differences were observed between CON and PLA (*p* = 0.321; *d =* 0.08). In conclusion, ingesting coffee providing 3 mg·kg^-1^ of caffeine increased salivary caffeine levels and improved 5 km cycling time trial performance in men and women by a similar magnitude.

## 1. Introduction

Caffeine ingestion is well established as an ergogenic aid for intermittent [1], endurance [2] and resistance exercise [3]. Furthermore, athletes regularly ingest caffeine in the form of coffee [4]. However, the majority of the research to date typically focuses on the ingestion of 3–8 mg·kg^-1^ of anhydrous caffeine, rather than the widely available source of caffeine, coffee [5], and predominantly in men [6], with only ten studies comparing men and women [7]. Consequently, there is a lack of research examining the ergogenic effect of caffeine on women [8], and in particular when receiving alternate forms of caffeine delivery, such as coffee [9].

Studies examining caffeine and exercise performance in women have produced varied results, with some reporting an ergogenic effect [10], whilst others suggest minimal benefit [11]. However, Skinner et al. [12] recently reported the ergogenic benefit of anhydrous caffeine was similar for men and women during endurance exercise. In contrast, Adan et al. [13] reported the effects of caffeine were greater in men, and a decaffeinated beverage produced greater effects in women, i.e., lesser somnolence and greater activation, defined as a combination of alertness and sleepiness. Additionally, Temple and Ziegler [14] reported men and women differ in their cardiovascular and subjective responses to caffeine, which may be mediated by changes in circulating steroid hormones, potentially affecting exercise performance. Therefore, it is of interest to investigate whether these differences in subjective and physiological responses are evident following coffee ingestion. Furthermore, there are equivocal findings with regard to the effect of menstrual cycle on caffeine metabolism (e.g., [15,16]), as well as oral contraceptive use being associated with decreased caffeine clearance [17].

There is growing evidence, albeit predominantly in men, that similar ergogenic benefits can be obtained from caffeine and coffee ingestion. Graham et al. [18] concluded that only anhydrous caffeine improved exercise performance when running at 85% maximal oxygen uptake and proposed that chlorogenic acids, and possibly other ingredients within the coffee, nullified the ergogenic benefits of caffeine. In contrast, Hodgson et al. [19] directly compared anhydrous caffeine to coffee during a cycling time trial and observed that both coffee and caffeine trials were beneficial to exercise performance, and by similar magnitudes. Similarly, Clarke et al. [20] reported that the ingestion of caffeinated coffee 60 minutes before a one-mile race markedly improved race performance compared with decaffeinated coffee and a placebo solution. McLellan and Bell [21] and Wiles et al. [22] also reported similar findings. In addition, Higgins et al. [5] concluded that coffee providing 3 mg·kg^-1^ of caffeine may be used as an alternative to anhydrous caffeine to improve endurance performance. Furthermore, recent reviews suggest that coffee ingestion should exert an ergogenic effect in most individuals [23] with a similar ergogenic effect of both caffeine and coffee, provided the caffeine dose is matched [24]. Therefore, the aim of the present study was to investigate whether sex differences exist with regard to 5 km cycling time trial performance and affective responses following the ingestion of caffeinated coffee.

## 2. Materials and Methods 

In a double-blind, Latin-square randomised, crossover, placebo-controlled design, 38 recreationally active participants (19 men: (Mean ± SD) age: 30 ± 5 years, height: 179 ± 6 cm, body mass: 81 ± 12 kg, maximal oxygen uptake (${\mathrm{VO}}_{2\max}$): 50 ± 9 mL·kg^-1^·min^-1^, body fat percentage: 26 ± 8%; physical activity (International Physical Activity Questionnaire (IPAQ) [25]): (median and interquartile range): 3276 (2282–4673) MET min·week^–1^; 19 women: age: 28 ± 6 years, height: 166 ± 7 cm, body mass: 72 ± 11 kg, ${\mathrm{VO}}_{2\max}$: 41 ± 9 mL·kg^-1^·min^-1^ body fat percentage: 33 ± 6%; physical activity (IPAQ [25]): (median and interquartile range): 3459 (2472–3813) MET min·week^–1^) completed a 5 km cycling time trial on a cycle ergometer following the ingestion of 0.09 g·kg^-1^ coffee (COF), a placebo (PLA) or control (no fluid ingestion; CON). Men had a higher body mass (*t(*36) = 2.621; *p* = 0.013; 95%CI: 2, 17; *d =* 0.86), fat-free mass (*t(*36) = 8.769; *p <* 0.001; 95%CI: 14, 22; *d =* 2.91) and ${\mathrm{VO}}_{2\max}$ (*t(*36)=2.985; *p* = 0.005; 95%CI: 3, 15; *d =* 0.97) but lower fat mass (*t(*36) = −8.631; *p <* 0.001; 95%CI: −17, −10; *d =* 2.81) and percentage body fat (*t(*36) = −4.156; *p <* 0.001; 95%CI: −12, −4; *d =* 1.36). There was no significant difference in physical activity (*t(*36) = 0.629; *p* = 0.533; 95%CI: −1042, 1981; *d =* 0.20), and hence, both men and women were defined as highly active [25]. All trials were conducted at the same time of day (09:00–12:00) and were consistent for each participant in order to minimise performance variation due to circadian factors. Women were required to be regularly taking a monophasic oral contraceptive pill for at least three months prior to the first trial. Both estrogen and oral contraceptive steroids appear to extend the half-life of caffeine, thereby prolonging its effects in the body [15], although this is unlikely to influence the performance of exercise of the nature used in the present study, due to its short duration. To control for possible differences during the oral contraceptive cycle, testing was performed during days 5–8 and 19–22 of the cycle, as no changes in energy metabolism or high-intensity intermittent exercise performance were reported between these time points [26]. All procedures were undertaken in accordance with the Declaration of Helsinki and approved by the institutional ethics committee (P70996). Participants were made fully aware of the exact procedures, including any risks and benefits of participation in the study before providing written informed consent. 

Before the experimental trials, participants completed a graded exercise test to exhaustion on a cycle ergometer (Wattbike Pro, Wattbike Ltd; Nottingham, UK) and a familiarisation of the 5 km time trial, performed on the same day. On arrival at the laboratory, nude body mass (Seca Alpha, Hamburg, Germany) and body composition (Tanita BC-418MA, Tanita Corporation, Tokyo, Japan) were measured after voiding. Participants adjusted the saddle and handlebars for comfort, and these adjustments remained consistent for all trials. After a 10 min warm-up, the test protocol began at 100 W and increased by 25 W every 2 min until volitional exhaustion. The warm-up consisted of a two-minute cycle at 70–80 rpm with air resistance set at “level 1” followed by an 8 min progressive build up to each participant’s peak power output achieved during the graded exercise test. The protocol consisted of 2 min of cycling at a self-selected power output, a further 2 min of self-paced cycling, including three 6 s maximal sprints and finishing with 3 min of cycling at a self-selected power output [27]. Breath-by-breath measurements were obtained throughout exercise using an online automated gas analysis system (Cortex Biophysik Meta- Max 3B) and averaged over the last 30 s in order to determine ${\mathrm{VO}}_{2\max}$. 

Habitual caffeine consumption was assessed using an adapted version of the Buhler et al. [28] caffeine consumption questionnaire. Some participants (four men and four women) were habitually high consumers of caffeine, with low habitual caffeine consumption defined as <300 mg·day^-1^ and >300 mg·day^-1^ defined as high [28] (Men: mean ± SD: 191 ± 118 mg·day-1 and range: 0–434 mg·day^−1^; Women: mean ± SD: 214 ± 158 mg·day^−1^ and range: 0–576 mg·day^−1^), although there were no significant differences between sexes (*t(*36) = −0.524; *p* = 0.604; 95%CI: −116, 68; *d =* 0.16). In addition, a 24 h dietary record was completed by each participant prior to the first trial; it was then analysed (Nutritics Research Edition v5.093, Nutritics, Dublin, Ireland), photocopied and returned to the participants, so that the same diet could be repeated for subsequent trials; daily energy: Men: 2160 ± 634 kcal vs. Women: 2043 ± 591 kcal (*t(*36) = 0.587; *p* = 0.561; 95%CI: −287, 520; *d =* 0.19); Carbohydrate: Men: 229 ± 87 g vs. Women: 222 ± 99 g (*t(*36) = 0.217; *p* = 0.829; 95%CI: −55, 68; *d =* 0.08); Protein: Men: 111 ± 45 g vs. Women: : 96 ± 27 g (*t(*36) = 1.211; *p* = 0.234; 95%CI: −10, 39; *d =* 0.40); Fat: Men: 83 ± 35 g vs. Women: 81 ± 31 g (*t(*36) = 0.129; *p* = 0.898; 95%CI: −20, 23; *d =* 0.06). Participants were also instructed to abstain from caffeine, alcohol and strenuous activity for at least 12 h, which along with diet was confirmed verbally prior to each trial. An overnight caffeine abstinence was employed as Pickering and Kiely [29] concluded that there appears to be no benefit from, and potentially negative consequences of, a short-term caffeine withdrawal period. Furthermore, a 3 mg·kg^−1^ dose of caffeine has been reported to significantly improve exercise performance irrespective of whether a 4 days withdrawal period was imposed on habitual caffeine users [30].

Nescafe original coffee (from the same batch; Nescafé Original, Nestlé, UK) was used in the coffee trials and dissolved in 300 mL of hot water (58 ± 3°C) and served in lidded cups. Participants were given a maximum of 10 min to consume the beverage, and the 60 min rest period started once the cup was emptied (Figure 1). The placebo trial was hot water of the same volume and temperature, and coffee flavour (Espresso Coffee Flavouring Compound, MSK Ingredients, UK) and colour (Brown Food Colouring, Lakeland, UK) was added to maintain treatment blinding and to ensure all trials tasted similar. A control condition was also conducted where the participants did not ingest any solutions. The coffee and placebo samples were analysed externally for their caffeine content (Laserchrom HPLC Laboratories Ltd, UK) using a high-performance liquid chromatography (HPLC) method. The coffee sample provided 35.1 mg of caffeine per 1 g of coffee, and the placebo contained no traces of caffeine. Based on this analysis, it was calculated that each participant consumed 0.09 g·kg^-1^ of coffee to achieve the 3 mg·kg^-1^ of caffeine required.

Following the rest period, participants performed the same ten min warm-up as described earlier. Following completion of the warm-up, there was a 60 s period where the participants were instructed to sit passively before a standardised countdown to initiate the time trial. The participants then performed a 5 km cycling time trial where they were instructed to complete the distance as fast as possible. During each time trial, participants had access to the distance remaining, and with the exception of verbal encouragement, no other information was provided. The gearing was self-selected by the participants on the Wattbike during the familiarisation trial and then replicated during each time trial. Bellinger and Minahan [27] reported that reliability of the Wattbike and test protocol makes it suitable for detecting “real” changes in performance, even where improvements may be small but still considered worthwhile. Trials were separated by a minimum period of 48 h to ensure complete recovery and caffeine washout.

Heart rate (HR) was measured throughout each time trial using a short-range telemetry HR transmitter strap (Polar RS400; Polar Electro Oy, Kempele, Finland). Saliva samples (minimum 0.5 mL by the passive drool technique) were obtained immediately before fluid ingestion to establish compliance with the washout period, one-hour post-ingestion prior to commencing the time trial and 60 s following the time trial. Participants were instructed to expectorate into a 20 mL plastic cup, and the sample was then transferred to a capped glass vial that was immediately placed in a −80°C freezer for subsequent analysis of caffeine concentration using a standard enzyme-linked immunoassay kit (Caffeine ELISA Kit; Creative Diagnostics, Shirley, USA). At the same time points, a capillary blood sample was drawn from the index finger for determination of blood glucose and lactate concentrations (Biosen C-line, EKF-diagnostic GmbH, Germany). Feeling scale (FS) [31], felt arousal scale (FAS) [32] and rating of perceived exertion (RPE) [33] were used to assess perceptual variables throughout trials. i.e., immediately before and after fluid ingestion and upon completion of every kilometre during the time trial (Figure 1).

### Data Analysis

A statistical power analysis was performed for sample size estimation based on the race time from a previous coffee-ingestion study [20]. As this study did not present correlation values between conditions, a conservative effect size value of 0.5 was used for the calculation. Consequently, an a priori power calculation suggests a sample size of 11 participants was deemed necessary to detect a difference between conditions given an estimated effect size of 0.44, a 1-β error probability of 0.95 and an α value of less than 0.05. However, based on the results of Fukuda et al. [34], where a moderate (partial eta squared ($\eta_{P}^{2}$)) sex–nutritional intervention interaction was identified, a sample size of 18 men and 18 women was required to identify any potential sex–coffee ingestion interaction. For all dependent variables, a mixed-design 2 (sex: men, women) × 3 (condition: caffeinated coffee, placebo, control) analysis of variance (ANOVA) was applied. Sex was computed as the independent factor, while condition was the within-subject factor. The data were analysed using IBM SPSS Statistics for Windows, Version 25.0 (Armonk, NY: IBM Corp.). In addition, due to the likely differences in baseline performance between men and women, an analysis of covariance (ANCOVA) was performed using the control condition as a covariant in order to further establish the ergogenic effect of coffee ingestion and whether sex differences exist. Furthermore, 95% confidence intervals (95%CI) and effect sizes, $\eta_{P}^{2}$, defined as trivial (<0.10), small (0.10–0.24), moderate (0.25–0.39) or large (≥0.40), and Cohen’s d, defined as trivial (<0.20), small (0.20–0.49), moderate (0.50–0.79) or large (≥0.80) according to the cut-offs suggested by Cohen [35], were also calculated. 

## 3. Results

There was a significant (F_2,61_=10.761; *p <* 0.001; $\eta_{P}^{2}$ = 0.23; Figure 2) improvement in 5 km time trial performance following the ingestion of coffee providing 3 mg·kg^-1^ of caffeine. Performance in the coffee trial (482 ± 51 s) was faster than the placebo (491 ± 53 s; *p* = 0.002; 95%CI: −15, −4; *d =* 0.17) and control (487 ± 52 s; *p* = 0.002; 95%CI: −9, −2; *d =* 0.10) trials. However, no differences were observed between control and placebo trials (*p* = 0.321; 95%CI: −9, 2, *d =* 0.08). Men (COF: 447 ± 32 s; PLA 457 ± 42 s; CON: 453 ± 34 s) were moderately faster than women (COF: 516 ± 42 s; PLA 526 ± 40 s; CON: 522 ± 43 s; *p <* 0.001; 95%CI: −94, −44; $\eta_{P}^{2}$ = 0.46). However, no significant sex and trial interaction was observed (F_2,61_=0.002; *p* = 0.998; $\eta_{P}^{2}$ = 0.00), with men and women both improving by approximately 9 and 6 s following coffee ingestion compared with placebo and control, respectively. The time trial performance equated to mean power outputs of overall: COF: 219 ± 61 W; PLA: 210 ± 61 W; CON: 212 ± 60 W; men: COF: 262 ± 52 W; PLA: 252 ± 58 W; CON: 254 ± 52 W; women: COF: 175 ± 38 W; PLA: 172 ± 35 W; CON: 175 ± 38 W. Furthermore, due to differences in baseline performance between men and women, an ANCOVA was performed using the control condition as a covariant, which demonstrated no significant difference in the improved performance (COF *vs*. PLA) following the ingestion of coffee between men and women (F_1,35_ = 0.631; *p* = 0.432; $\eta_{P}^{2}$ = 0.02), and as such, both improved by a similar magnitude. In addition, when assessed as a covariate, there was no significant effect of habitual caffeine consumption (F_1,32_ = 0.317; *p* = 0.577; $\eta_{P}^{2}$= 0.01) or fat-free mass (F_1,32_ = 0.191; *p* = 0.673; $\eta_{P}^{2}$ = 0.01) on the ergogenic effect of coffee consumption. 

A significant trial and time interaction for salivary caffeine levels was observed (F_4,144_ = 108.452; *p <* 0.001; $\eta_{P}^{2}$ = 0.75; Figure 3) with an increase (7.26 ± 2.08 μg·mL^−1^, 95%CI: 6.60, 7.91, *d =* 4.55) in salivary caffeine levels between baseline and post-drink following the ingestion of coffee. In contrast, only moderate and small increases were observed during CON (0.56 ± 1.36 μg·mL^−1^, 95%CI: 0.13, 0.99, *d* = 0.55) and PLA (0.45 ± 1.44 μg·mL^−1^, 95%CI: 0.00, 0.90, *d =* 0.44). Furthermore, no significant difference in salivary caffeine levels was observed between men and women (F_1,36_ = 0.022; *p* = 0.938; $\eta_{P}^{2}$ = 0.00).

A trial and time interaction for blood lactate was observed (F_3,112_ = 5.664; *p* = 0.001; $\eta_{P}^{2}$= 0.14; Table 1) with values higher at the completion of COF compared with CON (*p <* 0.001; 95%CI: 0.83, 1.99; *d =* 0.55) and PLA (*p* = 0.003; 95%CI: 0.44, 1.76; *d =* 0.41). Furthermore, no significant differences between CON and PLA were observed (*p* = 0.265; 95%CI: −0.22, 0.84; *d =* 0.11). Blood lactate values were higher in men compared with women (*p* = 0.001; 95%CI: 0.50, 1.73; $\eta_{P}^{2}$ = 0.27), although no significant trial and sex interaction was observed (F_2,72_ = 0.898; *p* = 0.412; $\eta_{P}^{2}$ = 0.02). Similarly, no significant main effect of heart rate was observed between trials (F_2,68_ = 3.315; *p* = 0.042; $\eta_{P}^{2}$ = 0.09; Table 2), with trivial to small higher heart rate values observed during COF compared with CON (*p* = 0.158; 95%CI: −1, 7; *d =* 0.21) and PLA (*p* = 0.041; 95%CI: 0, 5; *d =* 0.16). However, the magnitude was similar to the difference between CON and PLA (*p* = 1.00; 95%CI: −3, 3; *d =* 0.07). Furthermore, there were no significant differences in blood glucose concentrations between trials (F_2,56_ = 2.739; *p* = 0.085; $\eta_{P}^{2}$ = 0.07) and sexes (F_1,36_ = 0.221; *p* = 0.641; $\eta_{P}^{2}$ = 0.01; Table 1) observed. 

No significant differences between trials were observed for FAS (F_2,70_ = 0.521; *p* = 0.596; $\eta_{P}^{2}$ = 0.02; Table 2) and FS (F_2,72_ = 1.227; *p* = 0.299; $\eta_{P}^{2}$ = 0.03; Table 2), with no significant differences between men and women (FAS: F_1,35_ = 0.496; *p* = 0.486; $\eta_{P}^{2}$ = 0.01; FS: F_1,36_ = 2.671; *p* = 0.111; $\eta_{P}^{2}$ = 0.07). Similarly, no significant differences in RPE (Table 2) between trials (F_2,72_ = 0.677; *p* = 0.512; $\eta_{P}^{2}$ = 0.02) and sexes (F_1,36_ = 0.094; *p* = 0.761; $\eta_{P}^{2}$ = 0.00) were observed. 

## 4. Discussion

The aim of the present study was to investigate the effect of coffee ingestion on 5 km cycling time trial performance and to establish whether sex differences exist with regard to time trial performance and affective responses. Ingesting coffee providing 3 mg·kg^-1^ of caffeine improved 5 km cycling time trial performance in men and women, and by a similar magnitude. However, only trivial differences in the affective responses were observed between trials and sexes.

Following the ingestion of caffeinated coffee, 5 km cycling time trial performance was improved by 1.9% (Men: 2.1%; Women: 1.8%) and 1.2% (Men: 1.2%; Women: 1.1%) compared with PLA and CON, respectively. The ergogenic benefits of caffeine ingestion are well documented, e.g., [1,2]. Furthermore, Skinner et al. [12] recently reported that caffeine ingestion improved the time taken to complete a set amount of work (75% of peak sustainable power output) in men (4.6%) and women (4.3%) by similar magnitudes. However, fewer studies have documented the effects of coffee ingestion, especially comparing men and women. In support of the findings of the present study, Clarke et al. [20] reported that the ingestion of caffeinated coffee improved one-mile race time by 1.9% compared with a placebo. In addition, Wiles et al. [22] reported that the ingestion of 3 g of caffeinated coffee, containing approximately 150–200 mg of caffeine, improved 1500 m treadmill running performance by 4.2 s (1.4%) when compared with decaffeinated coffee. Similarly, Hodgson et al. [19] demonstrated that coffee and caffeine ingestion improved time trial performance by approximately 5%. Furthermore, Trexler et al. [36] observed that caffeine and coffee ingestion improved total work performed during repeated sprints (95%CI: 40, 219 joules). In contrast to the present study, Graham et al. [18] previously suggested that the bioactive compounds in coffee, such as chlorogenic acids, may attenuate the ergogenic effect of caffeine. Other contributing factors include the type of coffee and brewing method that impact on the amount of caffeine and chlorogenic acids present [37]. Furthermore, inter-individual variation in caffeine liver metabolism and pharmacodynamic and pharmacokinetic polymorphisms which have been linked to variation in response to caffeine [24] could also play a role. However, more recent studies [19,20,36] have reported both coffee and caffeine ingestion yield similar benefits for exercise performance. Therefore, the present study adds to the growing body of evidence highlighting the ergogenic benefit of coffee ingestion, and that in terms of exercise performance, men and women respond similarly to coffee.

Whilst there were no significant differences in FAS between the trials, FAS did significantly increase over time throughout all trials. These increases in FAS have been demonstrated in previous studies that have employed increasing levels of high-effort exercise [38]. Furthermore, all trials in the present study demonstrated similar decreases in FS throughout exercise [39]. Overall, these findings are similar to other studies that have measured the effects of high-intensity exercise, in that FS ratings decrease from a positive pre-exercise valance, towards 0 (neutral affect) but were not negative overall [40]. Furthermore, the observations of the present study are consistent with those of Welch et al. [39], who observed that FS ratings declined continuously throughout maximal effort cycling. Ekkekakis and Lind [41] suggest that when increasing exercise intensity initiates the transition from aerobic to anaerobic metabolism, there is a significant decline in affective valence and an increase in activation (FAS) as participants feel they are approaching exhaustion. Possibly due to the effort required to produce an improved time trial performance, evidenced by RPE, heart rate and lactate levels, coffee ingestion did not alleviate this phenomenon, and hence, FAS and FS were similar for COF, PLA and CON. 

Although the primary purpose of this study was not to determine the mechanism of coffee action, it is worth speculating on the underlying mechanism. The metabolism of caffeine is largely dependent on the *CYP1A2* isoform of cytochrome P450, with some evidence of downstream enzymatic differences between sexes [42], although this does not translate into altered exercise performance [12]. Adenosine antagonism, enhanced motor unit recruitment and reduced perception of pain and exertion have been proposed to explain the effects of caffeine supplementation on sport performance [24]. However, since caffeine interacts with many tissues, it is difficult to independently investigate its effects on the central and peripheral nervous systems and metabolism [43]. When specifically examining exercise of the nature in the present study, the primary mechanisms by which caffeine exerts its ergogenic effects are considered to arise from the antagonism of adenosine receptors, leading to an increase in neurotransmitter release and motor unit firing rates, pain suppression, reduced fatigue and improved neuromuscular performance [44]. Furthermore, Meeusen et al. [45] suggested that both motor effects and motivational aspects are influenced when adenosine receptors are blocked through caffeine, creating a greater dopaminergic drive, and thus enhancing 5 km performance. Another possible mechanism through which caffeine may improve performance is by increasing the secretion of β-endorphins [46], which may, at least partially, explain the mechanism by which caffeine attenuates pain sensation and rating of perceived exertion during exercise, thereby decreasing perceptions of effort and/or improving performance where maximal effort is required, as observed in the present study. Furthermore, in the present study, despite blood lactate concentration being higher at the completion of the COF trial, no differences in heart rate were observed, although these values were comparable with those of previous studies [18,19,22], suggesting that it is probable that coffee supplementation enhances physical performance via a combined effect on both the central and peripheral systems. 

The present study is not without limitations. Whilst the use of a 24 h dietary record to ensure pretrial standardisation is an acceptable method and a good reflection of standard nutritional practice, there may be some concern about its accuracy. A possible limitation is that caffeine consumption was restricted for only 12 hours. However, Irwin et al. [30] concluded that acute caffeine supplementation positively effects exercise performance and provides an ergogenic benefit in regular caffeine users regardless of any withdrawal period. A further potential limitation is the wide range of the participants’ habitual caffeine intake, although Gonçalves et al. [47] recently demonstrated that during ~30 min cycling time trials, performance was not influenced by habitual caffeine consumption. The beverages were administered in a double-blind manner, and the effectiveness of the blinding was examined using Bang’s blinding index (BBI [48]), where 1 represents a complete lack of blinding, 0 being consistent with perfect blinding and –1 indicating opposite guessing, which may be related to unblinding. One additional limitation might be that 27 out of 38 participants correctly identified the coffee trial. While the placebo was identified by random chance post-exercise (Mean and 95% CI: 0.04 (–0.04, 0.11)), the correct identification of coffee cannot be attributed solely to chance (Mean and 95% CI: 0.46 (0.38, 0.54)). The correct identification of coffee post-exercise is likely due to perceptions of “better overall feeling” and “more energy”, as reported previously [49]. Hence, there is the potential that the participants’ expectancy may have influenced the time-trial performance. These findings highlight the necessity of assessing a participants’ perception of what they have ingested in order to distinguish the true effect of caffeine from its placebo effect [50]. Finally, it would have been useful to assess affective responses post-exercise to check for (1) an affective rebound and (2) if that rebound differed between conditions.

## 5. Conclusions

Ingesting coffee providing 3 mg·kg^-1^ of caffeine increased salivary caffeine levels and improved 5 km cycling time trial performance in men and women by a similar magnitude, which suggests that recreationally active men and women respond similarly, and positively, to coffee ingestion prior to exercise. Furthermore, these results indicate that coffee ingestion may be a practical source of caffeine prior to a 5 km cycling time trial for men and women.

## Figures and Tables

**Figure 1 nutrients-11-02575-f001:**
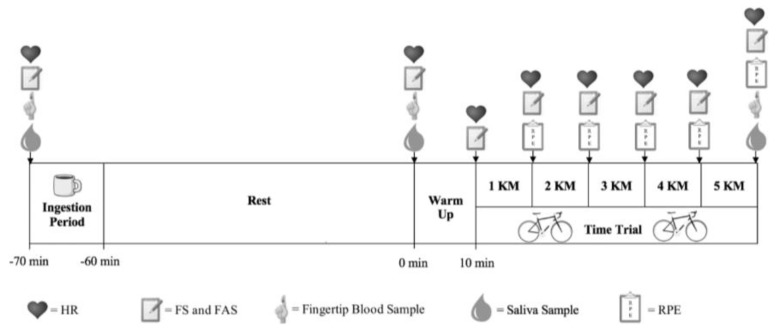
Schematic of the exercise protocol.

**Figure 2 nutrients-11-02575-f002:**
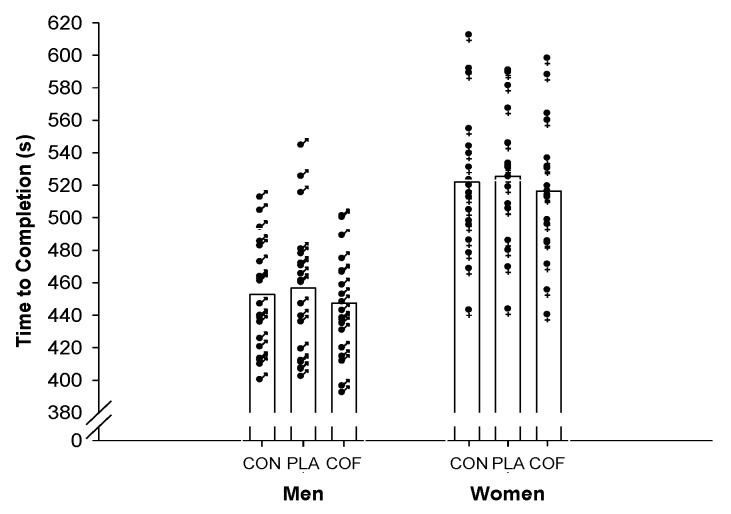
Mean (represented by the bar) and individual men (♂) and women (♀) 5 km cycling time trial completion.

**Figure 3 nutrients-11-02575-f003:**
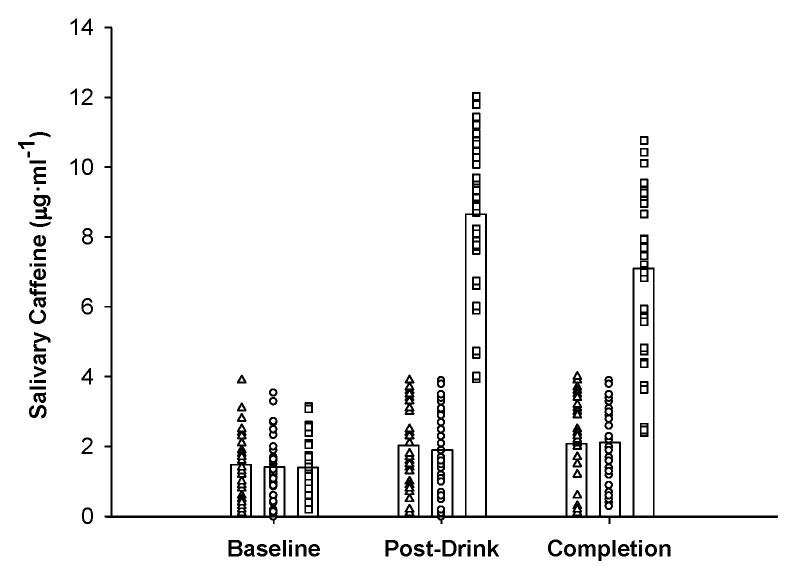
Mean and individual salivary caffeine responses during control (CON) (△), placebo (PLA) (○) and coffee (COF) (□) during the 5 km cycling time trial.

**Table 1 nutrients-11-02575-t001:** Mean ± SD lactate and glucose concentrations during the 5 km cycling time trial (*n* = 38).

	Pre-Drink	Post-Drink	Completion
Blood Lactate (mmol·l^-1^)			
CON	2.2 ± 1.2	2.3 ± 1.8	11.9 ± 2.5
PLA	2.0 ± 0.9	2.2 ± 1.0	12.2 ± 2.8
COF	2.1 ± 1.0	2.4 ± 1.1	13.3 ± 2.6
Blood Glucose (mmol·l^-1^)			
CON	4.7 ± 1.4	4.5 ± 1.0	4.6 ± 1.0
PLA	4.4 ± 0.7	4.3 ± 0.8	4.6 ± 1.0
COF	4.4 ± 1.0	4.3 ± 0.6	5.1 ± 1.0

**Table 2 nutrients-11-02575-t002:** Mean ± SD heart rate, rating of perceived exertion (RPE), feeling scale and felt arousal during the 5 km cycling time trial (*n* = 38).

	Pre-Drink	Post-Drink	1 km	2 km	3 km	4 km	5 km
Heart Rate (beats·min^-1^)							
CON	72 ± 15	71 ± 14	139 ± 18	159 ± 15	165 ± 13	169 ± 12	175 ± 11
PLA	72 ± 13	70 ± 13	138 ± 17	158 ± 16	164 ± 14	168 ± 12	173 ± 11
COF	73 ± 15	71 ± 14	140 ± 17	162 ± 15	168 ± 13	173 ± 12	178 ± 10
RPE							
CON			13 ± 2	14 ± 2	15 ± 2	16 ± 2	17 ± 2
PLA			13 ± 2	14 ± 2	15 ± 2	16 ± 2	17 ± 2
COF			13 ± 1	14 ± 2	15 ± 2	16 ± 2	17 ± 2
Feeling Scale							
CON	1.6 ± 2.0	2.1 ± 1.7	1.5 ± 1.7	1.1 ± 1.9	0.7 ± 2.2	0.4 ± 2.4	0.5 ± 2.4
PLA	2.1 ± 1.7	2.3 ± 1.4	1.9 ± 1.4	1.5 ± 1.7	1.0 ± 2.2	0.5 ± 2.5	0.5 ± 2.8
COF	1.8 ± 1.8	2.5 ± 1.3	1.8 ± 1.3	1.5 ± 1.5	1.0 ± 1.7	0.6 ± 1.9	0.4 ± 2.6
Felt Arousal							
CON	2.4 ± 1.1	2.3 ± 1.2	2.8 ± 0.9	3.0 ± 0.9	3.3 ± 1	3.4 ± 1.2	3.7 ± 1.1
PLA	2.3 ± 1.0	2.6 ± 0.9	3.0 ± 0.8	3.2 ± 0.9	3.2 ± 1	3.4 ± 1.1	3.8 ± 1.0
COF	2.3 ± 0.9	2.8 ± 1.1	3.0 ± 0.8	3.2 ± 0.8	3.4 ± 1	3.6 ± 1.1	3.8 ± 1.3

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
