# Peer review of "Coffee Ingestion Improves 5 km Cycling Performance in Men and Women by a Similar Magnitude"

_nutrients, 2019, doi:10.3390/nu11112575_

Round 1

Reviewer 1 Report

Artículo: La ingestión de café mejora el rendimiento de 5 km en hombres y mujeres en una magnitud similar.

Comparan la diferencia de rendimiento para la prueba de ciclismo de 5 km. Dosis 3mg · kg-1. comparación entre hombres y mujeres

La introducción está muy actualizada y es clara.

Métodos

Tengo algunas dudas, primero, la población es prácticamente sedentaria, sin familiarizarse con el cicloergómetro . Los porcentajes de grasa corporal total son muy altos, esto puede conducir a una sobreestimación de la dosis de cafeína según el peso corporal. Creo que sería ideal calcular la dosis de cafeína en función del porcentaje de masa corporal magra.

Los participantes con una alta ingesta de café pueden tener una menor excitación con la dosis proporcionada debido a la tolerancia. ¿Se introdujo como una covariable en el análisis estadístico?

La estandarización restante parece muy bien hecha.

¿Por qué no se controló la cadencia de pedaleo en el protocolo de prueba de esfuerzo máximo?

Salivary caffeine values raise some questions. After 12 hours of non-intake, the bioavailability of caffeine should be non-existent. Figure 3 can confuse the reader who expect this values to be zero: It should never last more than 14 hours. It would have been better to eliminate any chance of caffeine to be present in saliva and blood.

Comments:

Perform the analysis taking into account the body composition of the participants. Perform the analysis taking into account who are regular coffee consumers. Explain the reason why coffee consumption was not restricted for more than 12 hours. With a 24-hour rest, differences in caffeine in intragroup saliva would have been avoided. Defina qué es ser recreacionalmente activo para un sujeto (número de sesiones por semana y duración de la sesión). También si esta actividad es ciclismo o cuál / s.

Author Response

We thank the reviewers for taking time to review our manuscript and their positive comments, alongside constructive suggestions, which have improved the quality of the manuscript. We have highlighted any alterations in red font, and provided a summary below. Where the line numbers have changed, we had included the new line numbers in square brackets for ease of identifying any changes.

Reviewer 1

Article: Ingestion of coffee improves the yield of 5 km in men and women in a similar magnitude.

They compare the performance difference for the 5km cycling test. Dose 3mg · kg-1. comparison between men and women

The introduction is very updated and clear.

Thank you for this positive comment.

Methods

I have some doubts, first, the population is practically sedentary, without becoming familiar with the cycle ergometer. The percentages of total body fat are very high, this can lead to an overestimation of the dose of caffeine according to body weight. I think it would be ideal to calculate the dose of caffeine based on the percentage of lean body mass.

Participants with a high coffee intake may have less excitement with the dose provided due to tolerance. Was it introduced as a covariate in the statistical analysis?

The remaining standardization seems very well done.

Thank you for these positive comments.

Why was the pedaling rate not controlled in the maximum stress test protocol?

Due to the fixed resistance of the WattBike, power and speed were altered by cadence. During the time trial, participants were instructed to complete the 5 km as fast as possible. This has now been cited on [Lines 141-142].

Salivary caffeine values raise some questions. After 12 hours of non-intake, the bioavailability of caffeine should be non-existent. Figure 3 can confuse the reader who expect this values to be zero: It should never last more than 14 hours. It would have been better to eliminate any chance of caffeine to be present in saliva and blood.

Perform the analysis taking into account the body composition of the participants. Perform the analysis taking into account who are regular coffee consumers.

When assessed as a covariate, there was no significant effect of habitual caffeine consumption (F1,32=0.317; P=0.577; =0.01) or fat free mass (F1,32=0.191; P=0.673; =0.01) on the ergogenic effect of coffee consumption. [Lines 200-202].

Explain the reason why coffee consumption was not restricted for more than 12 hours. With a 24-hour rest, differences in caffeine in intragroup saliva would have been avoided.

Pickering and Kiely (2019) concluded that there appears to be no benefit from, and potentially negative consequences of, a short-term, pre-competition caffeine withdrawal period. Furthermore, a 3 mg·kg−1 dose of caffeine significantly improves exercise performance irrespective of whether a 4-day withdrawal period is imposed on habitual caffeine users (Irwin et al., 2011). [Lines 117-121]

Define what it is to be recreationally active for a subject (number of sessions per week and duration of the session). Also if this activity is cycling or which one / s.

Physical activity has now been included; Median and interquartile range: men: 3276 (2282 – 4673) MET min·week–1; women: 3459 (2472 – 3813) MET min·week–1. There was no significant difference in physical activity (t(36)=0.629; P=0.533; 95%CI: -1042, 1981; d=0.20), and hence both men and women were defined as highly active. [Lines 66-69, and 74-76].

Reviewer 2 Report

General Comments:

This research examines the impact of sex on the ergogenic effects of caffeine delivered in the form of coffee.  carbohydrate availability has on metabolic adaptations to exercise training. Notwithstanding an unclear hypothesis, the authors report that coffee consumption improved 5-km time trial performance to a similar extent between males and females. The research topic is certainly one of relevance and high interest. However, it appears that the investigators did not sufficiently blind the treatment to ameliorate the potential of a placebo effect. Further, more details is needed about feeding status and time of day that the participants completed testing. Finally, the discussion should be strengthened considerably before this manuscript can be recommended for publication. There are also minor inconsistencies and editorial error throughout (i.e. 5 km vs. 5km)

Specific Comments:

Title:

Specify that the performance criterion was 5-km cycling performance

Abstract:

Ln 12 – please add volume of fluid

Introduction:

Ln 28 – be specific about how many reports there are on men vs. women

Ln 35 – what is meant by ‘greater activation’

Methods:

Please detail the training level/recent physical activity habits in this cohort.

The authors mention that the time of day that the experiments occurred were held constant. What was the range and average time of day.

If there was variability in the time of day across subjects, subject must have eaten prior to the experiment. Please detail the dietary standardization employed here. Were subjects post-absorptive? Also, how long did subjects actually abstain from caffeine. This had to have varied widely if subjects reported to the laboratory at different times of the day (12 hrs of caffeine abstinence would have been impacted by sleep, therefore leading to much longer caffeine abstinence. This has the potential to influence the response to caffeine, particularly among habituated subjects.

Please include more details about the impact of monophasic oral contraceptives on the sex hormone profile and potential impact on caffeine metabolism and consequential ergogenic response.

Results:

Please include the individual responses either with a line of identity on a scatter plot or by connecting the individuals in Figure 2.

Ln 171-172 – It is unclear what ‘respectively’ is referring to. Is it referring to the men/women or placebo/control?

Please report mean power output during the time trials.

Discussion

The overall discussion reads like a literature review and not a discussion about the current findings in the context of the literature. Additionally, the relevance of the mechanism of coffee consumption paragraph is questionable, particularly with very little link to the current findings.

The fact that most subjects correctly identified the placebo trial, thereby introducing the possibility of a nocebo effect in addition to a placebo effect is a serious limitation that could entirely explain the ergogenic response to coffee consumption. This warrants more attention in the discussion.

Author Response

We thank the reviewers for taking time to review our manuscript and their positive comments, alongside constructive suggestions, which have improved the quality of the manuscript. We have highlighted any alterations in red font, and provided a summary below. Where the line numbers have changed, we had included the new line numbers in square brackets for ease of identifying any changes.

Reviewer 2

This research examines the impact of sex on the ergogenic effects of caffeine delivered in the form of coffee.  carbohydrate availability has on metabolic adaptations to exercise training. Notwithstanding an unclear hypothesis, the authors report that coffee consumption improved 5-km time trial performance to a similar extent between males and females. The research topic is certainly one of relevance and high interest. However, it appears that the investigators did not sufficiently blind the treatment to ameliorate the potential of a placebo effect. Further, more details is needed about feeding status and time of day that the participants completed testing. Finally, the discussion should be strengthened considerably before this manuscript can be recommended for publication. There are also minor inconsistencies and editorial error throughout (i.e. 5 km vs. 5km)

Specific Comments:

Title: Specify that the performance criterion was 5-km cycling performance

We appreciate the importance of stating the mode of exercise and have consequently added ‘cycling’ to the title.

Abstract:

Ln 12 – please add volume of fluid

Thank you for this suggestion, this has now been added [Line 13]

Introduction:

Ln 28 – be specific about how many reports there are on men vs. women

We recognise that this is an important consideration “Only ten studies have previously compared the ergogenic effect of caffeine between men and women (Mielgo-Ayuso et al., 2019).” [Line 29].

Ln 35 – what is meant by ‘greater activation’

A definition of activation has been included - a combination of alertness and sleepiness [Line 37].

Methods:

Please detail the training level/recent physical activity habits in this cohort.

We understand that this is an important point, and as such, the following has been added “Median and interquartile range: men: 3276 (2282 – 4673) MET min·week–1; women: 3459 (2472 – 3813) MET min·week–1. There was no significant difference in physical activity (t(36)=0.629; P=0.533; 95%CI: -1042, 1981; d=0.20), and hence both men and women were defined as highly active.” [Lines 66-69, and 74-76].

The authors mention that the time of day that the experiments occurred were held constant. What was the range and average time of day.

All trials were conducted at the same time of day (09:00 – 12:00), and was consistent for each participant in order to minimise performance variation due to circadian factors. [Lines 76-77].

If there was variability in the time of day across subjects, subject must have eaten prior to the experiment. Please detail the dietary standardization employed here. Were subjects post-absorptive? Also, how long did subjects actually abstain from caffeine. This had to have varied widely if subjects reported to the laboratory at different times of the day (12 hrs of caffeine abstinence would have been impacted by sleep, therefore leading to much longer caffeine abstinence. This has the potential to influence the response to caffeine, particularly among habituated subjects.

Participants were also instructed to abstain from caffeine, alcohol, and strenuous activity for at least 12 h, However, Pickering and Kiely (2019) concluded that there appears to be no benefit from, and potentially negative consequences of, a short-term, pre-competition caffeine withdrawal period. Furthermore, a 3 mg·kg−1 dose of caffeine significantly improves exercise performance irrespective of whether a 4-day withdrawal period is imposed on habitual caffeine users (Irwin et al., 2011). [Lines 117-121]

Please include more details about the impact of monophasic oral contraceptives on the sex hormone profile and potential impact on caffeine metabolism and consequential ergogenic response.

Both estrogen and oral contraceptive steroids (OCS) appear to extend the half-life of caffeine thereby prolonging its effects in the body (Lane et al., 1992), although this is unlikely to influence exercise of the nature in the present study due to its short duration. Furthermore, no changes in energy metabolism or high-intensity intermittent exercise performance have been reported between day 5 – 8 and 19 – 22 of the cycle (Lynch et al., 2001). [Lines 79-84]

Results:

Please include the individual responses either with a line of identity on a scatter plot or by connecting the individuals in Figure 2.

Thank you for this suggestion. We believe that the addition of lines will help the reader identify the individual responses.

Ln 171-172 – It is unclear what ‘respectively’ is referring to. Is it referring to the men/women or placebo/control?

This has been clarified with “Men and women both improved by approximately 9 seconds and 6 seconds following coffee ingestion compared with placebo and control, respectively.” [Lines 192-193]/

Please report mean power output during the time trials.

The time trial performance equated to power outputs of overall: COF: 219 ± 61 W; PLA: 210 ± 61 W; CON: 212 ± 60 W; Men: COF: 262 ± 52 W; PLA: 252 ± 58 W; CON: 254 ± 52 W; Women: COF: 175 ± 38 W; PLA: 172 ± 35 W; CON: 175 ± 38 W. [Lines 194-196].

Discussion

The overall discussion reads like a literature review and not a discussion about the current findings in the context of the literature. Additionally, the relevance of the mechanism of coffee consumption paragraph is questionable, particularly with very little link to the current findings.

We thank the reviewer for this suggestion and now believe that discussion is more focused on the findings of the present study. Furthermore, a clearer link has been made between the mechanisms and present study.

The fact that most subjects correctly identified the placebo trial, thereby introducing the possibility of a nocebo effect in addition to a placebo effect is a serious limitation that could entirely explain the ergogenic response to coffee consumption. This warrants more attention in the discussion.

The effectiveness of the blinding was examined using Bang’s blinding index (BBI; Bang et al., 2004), where 1 represents a complete lack of blinding, 0 being consistent with perfect blinding, and -1 indicating opposite guessing which may be related to unblinding. One additional limitation might be that 27 out of 38 participants correctly identified the coffee trial. While the placebo was identified by random chance post-exercise [Mean and 95% CI: 0.04 (-0.04, 0.11)], the correct identification of coffee cannot be attributed solely to chance [Mean and 95% CI: 0.46 (0.38, 0.54)]. The correct identification of coffee post-exercise is likely due to perceptions of “better overall feeling” and “more energy” as reported previously (Venier et al., 2019). Hence there is the potential that the participants’ expectancy may have influenced the time-trial performance and highlights the necessity of assessing a participants’ perception of what they have ingested in order to distinguish the true effect of a supplement from its placebo effect (Saunders et al., 2017). [Lines 336-346].

Treatment

Bang’s Blinding Index (Mean and 95% CI)

Coffee

0.46 (0.38, 0.54)

Placebo

0.04 (-0.04, 0.11)

Round 2

Reviewer 2 Report

Alterations to Figure 2 were not helpful to the reader. It would be much more clear with a line of identity, delta improvements relative to placebo, or grouping by sex so the individual responses are adjacent to each other. I will defer to the authors for the best way to display these. The bottom line is that it would be helpful for the reader to get some visual sense of the variability in the ergogenic response to caffeine.

Author Response

We thank the reviewer for their constructive comment. We have decided that grouping by sex so the individual responses are adjacent to each other is the clearest option for the reader.